# Fast, Accurate, and Simple Models for Tabular Data via Augmented Distillation

**Rasool Fakoor***
Amazon Web Services
fakoor@amazon.com

**Jonas Mueller***
Amazon Web Services
jonasmue@amazon.com

**Nick Erickson**
Amazon Web Services
neerick@amazon.com

**Pratik Chaudhari**
University of Pennsylvania
pratikac@seas.upenn.edu

**Alexander J. Smola**
Amazon Web Services
smola@amazon.com

## Abstract

Automated machine learning (AutoML) can produce complex model ensembles by stacking, bagging, and boosting many individual models like trees, deep networks, and nearest neighbor estimators. While highly accurate, the resulting predictors are *large, slow*, and *opaque* as compared to their constituents. To improve the deployment of AutoML on tabular data, we propose FAST-DAD to distill arbitrarily-complex ensemble predictors into individual models like boosted trees, random forests, and deep networks. At the heart of our approach is a data augmentation strategy based on Gibbs sampling from a self-attention pseudolikelihood estimator. Across 30 datasets spanning regression and binary/multiclass classification tasks, FAST-DAD distillation produces significantly better individual models than one obtains through standard training on the original data. Our individual distilled models are over $10\times$ faster and more accurate than ensemble predictors produced by AutoML tools like H2O/AutoSklearn.

## 1 Introduction

Modern AutoML tools provide good out-of-the-box accuracy on diverse datasets. This is often achieved through extensive model ensembling [1–3]. While the resultant predictors may generalize well, they can be large, slow, opaque, and expensive to deploy. Fig. 1 shows that the most accurate predictors can be 10,000 times slower than their constituent models.

Model distillation [4, 5] offers a way to compress the knowledge learnt by these complex models into simpler predictors with reduced inference-time and memory-usage that are also less opaque and easier to work with. In distillation, we train a simpler model (the *student*) to output similar predictions as those of a more complex model (the *teacher*). Here we use AutoML to create the most accurate possible teacher, typically an ensemble of many individual models via stacking, bagging, boosting, and weighted combinations [6]. Unfortunately, distillation typically comes with a sharp drop in accuracy.

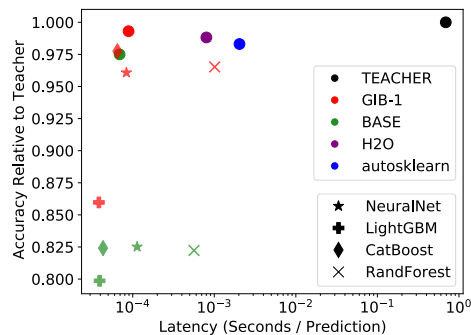

Figure 1: **Normalized test accuracy vs. speed of individual models and AutoML ensembles, averaged over all 30 datasets**. TEACHER denotes the performance of AutoGluon; H2O and autosklearn represent the respective AutoML tools. GIB-1 indicates the results of FAST-DAD after 1 round of Gibbs sampling. BASE denotes the student model fit on original data. GIB-1/BASE dots represent the model *Selected* (out of the 4 types) based on validation accuracy for each dataset.

---

Our paper mitigates this drop via FAST-DAD, a technique to produce **Fast**-and-accurate models via **D**istillation with **A**ugmented **D**ata. We apply FAST-DAD to large stack-ensemble predictors from AutoGluon [1] to produce individual models that are over $10{,}000\times$ faster than AutoGluon and over $10\times$ faster, yet still more accurate, than ensemble predictors produced by H2O-AutoML [7] and AutoSklearn [2].

**Motivation**. A key issue in distillation is that the quality of the student is largely determined by the amount of available training data. While standard distillation confers smoothing benefits (where the teacher may provide higher-quality prediction targets to the student [5, 8]), it incurs a student-teacher statistical approximation-error of similar magnitude as when training directly on original labeled dataset. By increasing the amount of data available for distillation, one can improve the student's approximation of the teacher and hence the student's accuracy on test data (assuming that the teacher achieves superior generalization error than fitting the student model directly to the original data). The extra data need not be labeled; one may use the teacher to label it. This enables the use of density estimation techniques to learn the distribution of the training data and draw samples of unlabeled data. In fact, we need not even learn the full joint distribution but simply learn how to *draw approximate samples* from it. We show that the statistical error in these new samples can be traded off against the reduction in variance from fitting the student to a larger dataset. Our resultant student models are almost as accurate as the teacher while being far more efficient/lightweight.

The contributions of this paper are as follows:

1. We present model-agnostic distillation that works across many types of teacher/student models and various supervised learning problems (binary/multiclass classification, regression). This is in contrast to problem and architecture-specific distillation techniques [4, 5, 9, 10].
2. We introduce a maximum pseudolikelihood model for tabular data that uses self-attention across covariates to simultaneously learn all of their conditional distributions.
3. We propose Gibbs sampling based on these conditional estimates to efficiently augment the dataset used in distillation. Our approach avoids estimating multivariate features' joint distribution, and enables control over sample-quality and diversity of the augmented dataset.
4. We report a comprehensive distillation benchmark for tabular data which studies 5 distillation strategies with 4 different types of student models over 30 regression/classification datasets.

Although our techniques can be adapted to other modalities, we focus on tabular data which has been under-explored in distillation despite its ubiquity in practical applications. Compared to typical data tables, vision and language datasets have far larger sample-sizes and with easily available data; data augmentation is thus not as critical for distillation as it is in the tabular setting.

## 2   Related Work

While distillation and model compression are popular in deep learning, existing work focuses primarily on vision, language and speech applications. Unlike the tabular settings we consider here, this prior work studies situations where: (a) unlabeled data is plentiful; (b) there are many more training examples than in typical data tables; (c) both teacher and student are neural networks; (d) the task is multiclass classification [5, 10–14].

For tabular data, Breiman and Shang [15] considered distilling models into single decision trees, but this often unacceptably harms accuracy. Recently, Vidal et al. [9] showed how to distill tree ensembles into a single tree without sacrificing accuracy, but their approach is restricted to tree student/teacher models. Like us, Bucilua et al. [4] considered distillation of large ensembles of heterogeneous models, pioneering the use of data augmentation in this process. Their work only considered binary classification problems with a neural network student model; multiclass classification is handled in a one-vs-all fashion which produces less-efficient students that maintain a model for every class. Liu et al. [16] suggest generative-adversarial networks can be used to produce better augmented data, but only conduct a small-scale distillation study where learners are random forest models.

## 3   From Function Approximation to Distillation

We first formalize distillation to quantify the role of the auxiliary data in this process. Consider a dataset $(X_n, Y_n)$ where $X_n = \left\{ x_i \in \mathcal{X} \subset \mathbb{R}^d \right\}_{i=1}^n$ are observations of some features sampled from distribution $p$, and $Y_n = \{ y_i \in \mathcal{Y} \}_{i=1}^n$ are their labels sampled from distribution $p_{y|x}$. The teacher

$f : \mathcal{X} \to \mathcal{Y}$ is some function learned e.g. via AutoML that achieves good generalization error:

$$R[f] := \mathbf{E}_{(x,y)\sim p}\left[\ell(f(x), y)\right]$$

where loss $\ell$ measures the error in individual predictions. Our goal is to find a model $g$ from a restricted class of functions $\mathcal{G} \subset \mathcal{L}^2(\mathcal{X})$ such that $R[g]$ is smaller than the generalization error of another model from this class produced via empirical risk minimization.

**Approximation**. Distillation seeks some student $g^*$ that is "close" to the teacher $f$. If $\|f - g^*\|_\infty \leqslant \epsilon$ over $\mathcal{X}$ and if the loss function $\ell$ is Lipschitz continuous in $f(x)$, then $g^*$ will be nearly as accurate as the teacher ($R[g^*] \approx R[f]$). Finding such a $g^*$ may however be impossible. For instance, a Fourier approximation of a step function will never converge uniformly but only pointwise. This is known as the *Gibbs phenomenon* [17]. Fortunately, $\ell_\infty$-convergence is not required: we only require convergence with regard to some distance function $d(f(x), g(x))$ averaged over $p$. Here $d$ is determined by the task-specific loss $\ell$. For instance, $\ell_2$-loss can be used for regression and the KL divergence between class-probability estimates from $f, g$ may be used in classification. Our goal during distillation is thus to minimize

$$D(f, g, p) = \mathbf{E}_{x \sim p}\left[d(f(x), g(x))\right]. \tag{1}$$

This is traditionally handled by minimizing its empirical counterpart [5]:

$$D_{\mathrm{emp}}(f, g, X_n) = \frac{1}{n}\sum_{i=1}^n d(f(x_i), g(x_i)). \tag{2}$$

**Rates of Convergence**. Since it is only an empirical average, minimizing $D_{\mathrm{emp}}$ over $g \in \mathcal{G}$ will give rise to an approximation error that can bounded, e.g. by uniform convergence bounds from statistical learning theory [18] as $O(\sqrt{V/n})$. Here $V$ denotes the complexity of the function class $\mathcal{G}$ and $n$ is the number of observations used for distillation. Note that we effectively pay twice for the statistical error due to sampling $(X_n, Y_n)$ from $p$. Once to learn $f$ and again while distilling $g^*$ from $f$ using the same samples.

There are a number of mechanisms to reduce the second error. If we had access to more unlabeled data, say $X'_m$ with $m \gg n$ drawn from $p$, we could reduce the statistical error due to distillation significantly (as empirically demonstrated in Fig. S2). While we usually cannot draw from $p$ for tabular data due to a lack of additional unlabeled examples (unlike say for images/text), we might be able to draw from a related distribution $q$ which is sufficiently close. In this case we can obtain a uniform convergence bound:

**Lemma 1 (Surrogate Approximation)** *Assume that the complexity of the function class $\mathcal{G}$ is bounded under $d(f(x), \cdot)$ and $d(f(x), g(x)) \leqslant 1$ for all $x \in \mathcal{X}$ and $g \in \mathcal{G}$. Then there exists a constant $V$ such that with probability at least $1 - \delta$ we have*

$$D(f, g^*, p) \leqslant D_{\mathrm{emp}}(f, g^*, X'_m) + \sqrt{(V - \log \delta)/m} + \|p - q\|_1. \tag{3}$$

*Here $X'_m$ are $m$ samples from $q$ and $g^* \in \mathcal{G}$ is chosen, e.g. to minimize $D_{\mathrm{emp}}(f, g, X'_m)$.*

**Proof** This follows directly from Hölder's inequality when applied to $D(f, g, p) - D(f, g, q) = \int l(f(x), g(x))(p(x) - q(x))dx \leqslant C\|p - q\|_1$. Next we apply uniform convergence bounds to the difference between $D_{\mathrm{emp}}(f, g, X'_m) - D(f, g, q)$. Using VC bounds [18] proves the claim. ■

The inequality (3) suggests a number of strategies when designing algorithms for distillation. Whenever $p$ and $q$ are similar in terms of the bias $\|p - q\|_1$ being small, we want to draw as much data as we can from $q$ to make the uniform convergence term vanish. However if $q$ is some sort of estimate, a nontrivial difference between $p$ and $q$ will usually exist in practice. In this case, we may trade off the variance reduction offered by extra augmented samples and the corresponding bias by drawing these samples from an intermediate distribution that lies *in between* the training data and the biased $q$.

## 4  FAST-DAD Distillation via Augmented Data

The augmentation distribution $q$ in Lemma 1 could be naively produced by applying density estimation to the data $X_n$, and then sampling from the learnt density. Unfortunately, multivariate density estimation and generative modeling are at least as difficult as the supervised learning problems

AutoML aims to solve [19]. It is however much easier to estimate $p(x^i|x^{-i})$, the univariate conditional of the feature $x^i$ given all the other features $x^{-i}$ in datum $x = (x^i, x^{-i})$. This suggests the following strategy which forms the crux of FAST-DAD:

1. For all features $i$: estimate conditional distribution $p(x^i|x^{-i})$ using the training data.
2. Use all training data $x \in X_n$ as initializations for a Gibbs sampler [20]. That is, use each $x \in X_n$ to generate an MCMC chain via: $\tilde{x}^i \sim p(x^i|x^{-i}), x^i \leftarrow \tilde{x}^i$.
3. Use the samples from all chains as additional data for distillation.

We next describe these steps in detail but first let us see why this strategy can generate good augmented data. If our conditional probability estimates $p(x^i|x^{-i})$ are accurate, the Gibbs sampler is guaranteed to converge to samples drawn from $p(x)$ regardless of the initialization [21]. In particular, initializing the sampler with data $x \in X_n$ ensures that it doesn't need time to 'burn-in'; it starts immediately with samples from the correct distribution. Even if $p(x^i|x^{-i})$ is inaccurate (inevitable for small $n$), the sample $\tilde{x}$ will not deviate too far from $p(x)$ after a small number of Gibbs sampling steps (low bias), whereas using $\tilde{x} \sim q$ with an inaccurate $q$ would produce disparate samples.

## 4.1 Maximum Pseudolikelihood Estimation via Self-Attention

A cumbersome aspect of the strategy outlined above is the need to model many conditional distributions $p(x^i|x^{-i})$ for different $i$. This would traditionally require many separate models. Here we instead propose a single self-attention architecture [22] with parameters $\theta$ that is trained to simultaneously estimate all conditionals via a pseudolikelihood objective [23]:

$$\hat{\theta} = \underset{\theta}{\arg\max} \frac{1}{n} \sum_{x \in X_n} \sum_{i=1}^{d} \log p(x^i \,|x^{-i}; \theta) \qquad (4)$$

For many models, maximum pseudolikelihood estimation produces asymptotically consistent parameter estimates, and often is more computationally tractable than optimizing the likelihood [23]. Our model takes as input $(x^1, \dots, x^d)$ and simultaneously estimates the conditional distributions $p(x^i|x^{-i}; \theta)$ for all features $i$ using a self-attention-based encoder. As in Transformers, each encoder layer consists of a multi-head self-attention mechanism and a feature-wise feedforward block [22]. Self-attention helps this model gather relevant information from $x^{-i}$ needed for modeling $x^i$.

Each conditional is parametrized as a mixture of Gaussians $p(x^i|x^{-i}; \theta) = \sum_{k=1}^{K} \lambda_k N(x^i; \mu_k, \sigma_k^2)$, where $\lambda_k, \mu_k, \sigma_k$ depend on $x^{-i}$ and are output by topmost layer of our encoder after processing $x^{-i}$. Categorical features are numerically represented using dequantization [24]. To condition on $x^{-i}$ in a mini-batch (with $i$ randomly selected per mini-batch), we mask the values of $x^i$ to omit all information about the corresponding feature value (as in [25]) and also mask all self-attention weights for input dimension $i$; this amounts to performing stochastic gradient descent on the objective in (4) across both samples and their individual features. We thus have an efficient way to compute any of these conditional distributions with one forward pass of the model. While this work utilizes self-attention, our proposed method can work with any efficient estimator of $p(x^i|x^{-i})$ for $i = 1, \dots, d$.

**Relation to other architectures**. Our approach can be seen as an extension of the mixture density network [26], which can model arbitrary conditional distributions, but not all conditionals simultaneously as enabled by our use of masked self-attention with the pseudolikelihood objective. It is also similar to TraDE [27]: however, their auto-regressive model requires imposing an arbitrary ordering of the features. Since self-attention is permutation-invariant [28], our pseudolikelihood model is desirably insensitive to the order in which features happen to be recorded as table columns. Our use of masked self-attention shares many similarities with BERT [25], where the goal is typically representation learning or text generation [29]. In contrast, our method is designed for data that lives in tables. We need to estimate the conditionals $p(x^i|x^{-i}; \theta)$ very precisely as they are used to sample continuous values; this is typically not necessary for text models.

## 4.2 Gibbs Sampling from the Learnt Conditionals

We adopt the following procedure to draw Gibbs samples $\tilde{x}$ to augment our training data: The sampler is initialized at some training example $x \in X_n$ and a random ordering of the features is selected (with different orderings used for different Gibbs chains started from different training examples). We cycle through the features and in each step replace the value of one feature in $\tilde{x}$, say $\tilde{x}^i$, using its

conditional distribution given all the other variables, i.e. $p(x^i \mid x^{-i}; \widehat{\theta})$. After every feature has been resampled, we say one *round* of Gibbs sampling is complete, and proceed onto the next round by randomly selecting a new feature-order to follow in subsequent Gibbs sampling steps.

A practical challenge in Gibbs sampling is that a poor choice of initialization may require many burn-in steps to produce reasonable samples. Suppose for the following discussion that our pseudolikelihood estimator and its learnt conditionals are accurate. We can use a strategy inspired by Contrastive Divergence [30] and initialize the sampler at $x \in X_n$ and take a few (often only one) Gibbs sampling steps. This strategy is effective; we need not wait for the sampler to burn in because it is initialized at (or close to) the true distribution itself. This is seen in Fig. 2 where we compare samples from the true distribution and Gibbs samples (taken with respect to conditional estimates from our self-attention network) starting from an arbitrary initialization vs. initialized at $X_n$.

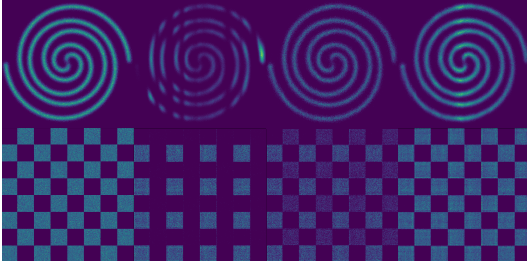

Figure 2: **Initialization of the Gibbs sampler.** From left to right: original training data, samples obtained from one round of Gibbs sampling with random initialization after fitting the self-attention network, samples obtained after multiple rounds of Gibbs sampling (10 for the spiral, 100 for the checkerboard density) with random initialization, and samples obtained from one Gibbs sampling round when initializing via $X_n$. The densities were generated from examples in Nash and Durkan [31].

For distillation, we expect this sampling strategy to produce better augmented data. The number of Gibbs sampling steps provides fine-grained control over the sample fidelity and diversity of the resulting dataset used in distillation. Recall that the student will be trained over $X_n \cup X'_m$ in practice. When our estimates of $p(x^i|x^{-i})$ are accurate, it is desirable to produce $X'_m$ only after a large number of Gibbs steps, as the bias in (3) will remain low and we would like to ensure the $X'_m$ are more statistically independent from $X_n$. With worse estimates of $p(x^i|x^{-i})$, it is better to produce $X'_m$ after only a few Gibbs steps to ensure lower bias in (3), but the lack of burn-in implies $X'_m$ are not independent of $X_n$ and may thus be less useful to the student during distillation. We dig deeper into this phenomenon (for the special case of $m = n$) in the following theorem.

**Theorem 2 (Refinement of Lemma 1)** *Under the assumptions of Lemma 1, suppose the student $g^*$ minimizes $D_{\mathrm{emp}}(f, g, X_n \cup X'_n)$ where $X'_n$ are $n$ samples drawn after $k$ steps of the Gibbs sampler initialized at $X_n$. Then there exist constants $V, c, \delta > 0$ such that with probability $\geqslant 1 - \delta$:*

$$D(f, g^*, p) \leqslant D_{\mathrm{emp}}(f, g^*, X_n \cup X'_n) + \sqrt{\frac{4V(c + \Delta_k) - \log \delta}{n}} + \Delta_k \qquad (5)$$

$\Delta_k = \|T_q^k p - p\|_{\mathrm{TV}}$ *is the total-variation norm between $p$ and $T_q^k p$ (the distribution of Gibbs samples after $k$ steps), where $q$ denotes the steady-state distribution of the Gibbs sampler.*

The proof (in Appendix D) is based on multi-task generalization bounds [32] and MCMC mixing rates [33]. Since $\Delta_k \to \|T_q^k p - q\|_{\mathrm{TV}}$ as $k \to \infty$, we should use Gibbs samples from a smaller number of steps when $q$ is inaccurate (e.g. if our pseudolikelihood estimator is fit to limited data).

## 4.3 Training the Student with Augmented Data

While previous distillation works focused only on particular tasks [4, 5], we consider the range of regression and classification tasks. Our overall approach is the same for each problem type:

1. Generate a set of augmented samples $X'_m = \{x'_k\}_{k=1,\ldots,m}$.
2. Feed the samples $X'_m$ as inputs to the teacher model to obtain predictions $Y'_m$, which are the predicted class probabilities in classification (rather than hard class labels), and predicted scalar values in regression.
3. Train each student model on the augmented dataset $(X_n, Y_n) \cup (X'_m, Y'_m)$.

In the final step, our student model is fit to a combination of both true labels from the data $y$ as well as as augmented labels $y'$ from the teacher, where $y'$ is of different form than $y$ in classification (predicted probabilities rather than predicted classes). For binary classification tasks, we employ the Brier score [34] as our loss function for all students, treating both the probabilities assigned to the

positive class by the teacher and the observed $\{0, 1\}$ labels as continuous regression targets for the student model. The same strategy was employed by Bucilua et al. [4] and it slightly outperformed our alternative multiclass-strategy in our binary classification experiments. We handle multiclass classification in a manner specific to different types of models, avoiding cumbersome students that maintain a separate model for each class (c.f. one-vs-all). Neural network students are trained using the cross-entropy loss which can be applied to soft labels as well. Random forest students can utilize multi-output decision trees [35] and thus be trained as native multi-output regressors against targets which are one-hot-encoded class labels in the real data and teacher-predicted probability vectors in the augmented data. Boosted tree models are similarly used to predict vectors with one dimension per class, which are then passed through a softmax transformation; the cross entropy loss is minimized via gradient boosting in this case.

## 5  Experiments

**Data**. We evaluate various methods on 30 datasets (Table S2) spanning regression tasks from the UCI ML Repository and binary/multi classification tasks from OpenML, which are included in popular deep learning and AutoML benchmarks [1, 36–40]. To facilitate comparisons on a meaningful scale across datasets, we evaluate methods on the provided test data based on either their accuracy in classification, or percentage of variation explained $(= R^2 \cdot 100)$ in regression. The training data are split into training/validation folds (90-10), and only the training fold is used for augmentation (validation data keep their original labels for use in model/hyper-parameter selection and early-stopping).

**Setup**. We adopt AutoGluon as our teacher as this system has demonstrated higher accuracy than most other AutoML frameworks and human data science teams [1]. AutoGluon is fit to each training dataset for up to 4 hours with the `auto_stack` option which boosts accuracy via extensive model ensembling (all other arguments left at their defaults). The most accurate ensembles produced by AutoGluon often contain over 100 individual models trained via a combination of multi-layer stacking with repeated 10-fold bagging and the use of multiple hyperparameter values [1]. Each model trained by AutoGluon is one of: (1) Neural Network (NN), (2) CatBoost, (3) LightGBM, (4) Random Forest (RF), (5) Extremely Randomized Trees, and (6) K-Nearest Neighbors.

We adopt the most accurate AutoGluon ensemble (on the validation data) as the teacher model. We use models of types (1)-(4) as students, since these are more efficient than the others and thus more appropriate for distillation. These are also some of the most popular types of models among today's data scientists [41]. We consider how well each individual type of model performs under different training strategies, as well as the overall performance achieved with each strategy after a model selection step in which the best individual model on the validation data (among all 4 types) is used for prediction on the test data. This *Selected* model reflects how machine learning is operationalized in practice. All candidate student models (as well as the BASE models) of each type share the same hyper-parameters and are expected to have similar size and inference latency.

### 5.1  Distillation Strategies

We compare our FAST-DAD Gibbs-augmented distillation technique with the following methods.

**TEACHER**: The model ensemble produced by AutoGluon fit to the training data. This is adopted as the teacher in all distillation strategies we consider.

**BASE**: Individual base models fit to the original training data (trained in the usual manner).

**KNOW**: *Knowledge distillation* proposed by Hinton et al. [5], in which we train each student model on the original training data, but with labels replaced by predicted probabilities from the teacher which are smoothed and nudged toward the original training labels (no data augmentation).

**MUNGE**: The technique proposed by Bucilua et al. [4] to produce augmented data for model distillation, where the augmented samples are intended to resemble the underlying feature distribution. MUNGE augmentation may be viewed as a few steps of Gibbs sampling, where $p(x^i | x^{-i})$ is estimated by first finding near neighbors of $x$ in the training data and subsequently sampling from the (smoothed) empirical distribution of their $i^{\text{th}}$ feature [43].

**HUNGE**: To see how useful the teacher's learned label-distributions are to the student, we apply a *hard* variant of MUNGE. Here we produce MUNGE-augmented samples that receive hard class

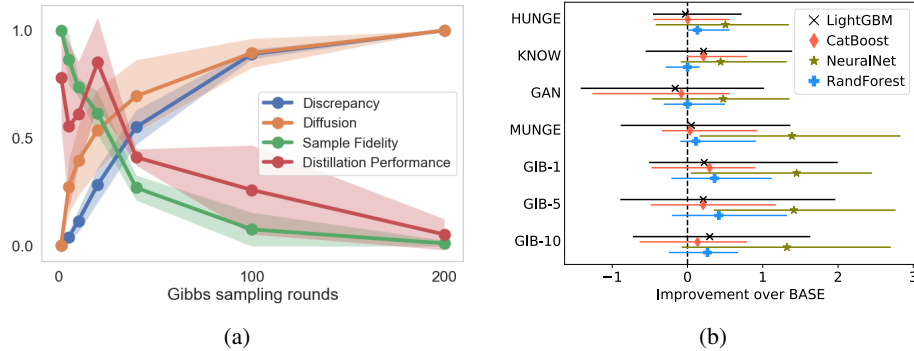

(a)                           (b)

Figure 3: **Fig. 3a Normalized metrics evaluated on samples from various Gibbs rounds (averaged across 3 datasets).** *Sample Fidelity* measures how well a random forest discriminator can distinguish between (held out) real and Gibbs-sampled data. *Diffusion* is the average Euclidean distance between each Gibbs sample and the datum from which its Markov chain was initialized. *Discrepancy* is the Maximum Mean Discrepancy [42] between the Gibbs samples and the training data; it measures both how well the samples approximate $p$ as well as how distinct they are from data $X_n$. *Distillation Performance* is the test accuracy of student models trained on the augmented data (averaged over our 4 model types). The diversity of the overall dataset used for distillation grows with increased discrepancy/diffusion, while this overall dataset more closely resembles the underlying data-generating distribution with increased sample fidelity (lower bias). The discriminator's accuracy ranges between $[0.49, 0.90]$ for these datasets. **Fig. 3b Percentage points improvement over the BASE model produced by each distillation method** for different model types (change in: accuracy for classification, explained variation for regression). As the improvements contain outliers/skewness, we show the median change across all datasets (dots) and the corresponding interquartile range (lines).

predictions from the teacher as their labels rather than the teacher's predicted probabilities that are otherwise the targets in all other distillation strategies (equivalent to MUNGE for regression).

**GAN**: The technique proposed by Xu et al. [44] for augmenting tabular data using conditional generative adversarial networks (GANs), which have been shown to produce higher-quality samples than other deep generative models. As in our approach, this GAN is trained on the training set and then used to generate augmented $x$ samples for the student model, whose labels are the predicted probabilities output by the teacher. Unlike our Gibbs sampling strategy, it is difficult to control how similar samples from the GAN should be to the training data.

We also run our Gibbs sampling data augmentation strategy generating samples after various numbers of Gibbs sampling rounds (for example, **GIB-5** indicates 5 rounds were used to produce the augmented data). Under each augmentation-based strategies, we add $m$ synthetic datapoints to the training set for the student, where $m = 10\times$ the number of original training samples (up to at most $10^6$).

## 5.2   Analysis of the Gibbs Sampler

To study the behavior of our Gibbs sampling procedure, we evaluate it on a number of different criteria (see Fig. 3a caption). Fig. 3a depicts how the distillation dataset's overall diversity increases with additional rounds of Gibbs sampling. Fortuitously, we do not require a large number of Gibbs sampling rounds to obtain the best distillation performance and can thus efficiently generate augmented data. Running the Gibbs sampling for longer is ill-advised as its stationary distribution appears to less closely approximate $p$ than intermediate samples from a partially burned-in chain; this is likely due to the fact that we have limited data to fit the self-attention network.

## 5.3   Performance of Distilled Models

Table 1 and Fig. 3b demonstrate that our Gibbs augmentation strategy produces far better resulting models than any of the other strategies. Table S3 shows the only datasets where Gibbs augmentation fails to produce better models than the BASE training strategy are those where the teacher ensemble fails to outperform the best individual BASE model (so little can be gained from distillation period). As expected according to Hinton et al. [5]: KNOW helps in classification but not regression, and HUNGE fares worse than MUNGE on multiclass problems where its augmented hard class-labels fail to provide students with the teacher's *dark knowledge*. As previously observed [4], MUNGE greatly

Table 1: **Average ranks/performance achieved by the *Selected* model** under each training strategy across the datasets from each prediction task. Performance is test accuracy for classification or percentage of variation explained for regression, and we list $p$-values for the one-sided test of whether each strategy $\geqslant$ BASE.

| Strategy | Rank | Accuracy | $p$ | Rank | Accuracy | $p$ | Rank | Accuracy | $p$ |
|---|---|---|---|---|---|---|---|---|---|
| BASE | 6.888 | 88.63 | - | 5.791 | 82.85 | - | 7.777 | 80.80 | - |
| HUNGE | 5.0 | 88.99 | 0.092 | 5.541 | 83.57 | 0.108 | 7.666 | 81.04 | 0.350 |
| KNOW | 6.555 | 88.49 | 0.712 | 5.25 | 83.89 | 0.072 | 5.555 | 81.39 | 0.275 |
| GAN | 6.666 | 88.65 | 0.450 | 6.708 | 83.17 | 0.250 | 6.055 | 82.26 | 0.069 |
| MUNGE | 5.444 | 88.88 | 0.209 | 5.083 | 83.72 | 0.126 | 4.333 | 82.80 | 0.007 |
| GIB-1 | 3.777 | **89.35** | 0.025 | **3.708** | **84.21** | **0.051** | **3.277** | **82.88** | **0.005** |
| GIB-5 | **3.333** | 89.25 | **0.004** | 5.375 | 84.04 | 0.098 | 3.388 | 82.76 | 0.010 |
| GIB-10 | 4.777 | 89.09 | 0.044 | 4.958 | 83.74 | 0.087 | 4.222 | 82.64 | 0.010 |
| TEACHER | 2.555 | 90.10 | 0.036 | 2.583 | 84.40 | 0.019 | 2.722 | 83.84 | 0.018 |
| | **Regression Problems** | | | **Binary Classification** | | | **Multiclass Classification** | | |

improves the performance of neural networks, but provides less benefits for the other model types than augmentation via our Gibbs sampler. Overparameterized deep networks tend to benefit from distillation more than the tree models in our experiments (although for numerous datasets distilled tree models are still *Selected* as the best model to predict with). While neural nets trained in the standard fashion are usually less accurate than trees for tabular data, FAST-DAD can boost their performance above that of trees, a goal other research has struggled to reach [45–49].

Figs. 1, 4 and S1 depict the (normalized/raw) accuracy and inference-latency of our distilled models (under the GIB-1 strategy which is superior to others), compared with both the teacher (AutoGluon ensemble), as well as ensembles produced by H2O-AutoML [7] and AutoSklearn [2], two popular AutoML frameworks that have been shown to outperform other AutoML tools [40, 50]. On average, the *Selected* individual model under standard training (BASE) would be outperformed by these AutoML ensembles, but surprisingly, our distillation approach produces *Selected* individual models that are both more accurate and over $10\times$ more efficient than H2O and AutoSklearn. In multiclass classification, our distillation approach also confers significant accuracy gains over standard training. The resulting individual *Selected* models come close to matching the accuracy of H2O/AutoSklearn while offering much lower latency, but gains may be limited since the AutoGluon teacher appears only marginally more accurate than H2O/AutoSklearn in these multiclass problems.

# 6 Discussion

Our goal in this paper is to build small, fast models that can bootstrap off large, ensemble-based AutoML predictors via model distillation to perform better than they would if directly fit to the original data. A key challenge is the data to train the student are limited. We propose to estimate the conditional distributions of all features via maximum pseudolikelihood with masked self-attention, and use Gibbs sampling to augment the data available to the student. Our strategy neatly suits this application because it: (i) avoids multivariate density estimation (pseudolikelihood only involves univariate conditionals), (ii) does not require separate models for each conditional (one self-attention model simultaneously computes all conditionals), (iii) is faster than usual MCMC (by initializing the Gibbs sampler at the training data), (iv) allows control over the quality and diversity of the resulting augmented dataset (we can select samples from specific Gibbs rounds unlike from, say, a GAN). We used the high-accuracy ensembles produced by AutoGluon's AutoML [1] to improve standard boosted trees, random forests, and neural networks via distillation.

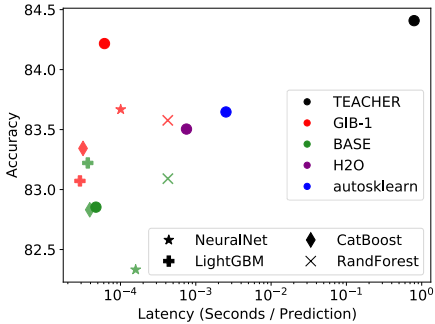

Figure 4: **Raw test accuracy vs. speed of individual models and AutoML ensembles, averaged over binary classification datasets**. TEACHER denotes the performance of AutoGluon; H2O and autosklearn represent the respective AutoML tools. GIB-1 indicates the results of FAST-DAD after 1 round of Gibbs sampling. BASE denotes the student model fit on the original data. GIB-1/BASE dots represent the *Selected* model.

## Broader Impact Statement

This work will potentially impact the community in two main ways. Our proposal to use high-accuracy AutoML ensembles followed by model distillation allows practitioners to deploy their favorite models, but obtain significantly better accuracy than they could fitting these models directly to their data in the standard fashion. Our work thus helps realize AutoML's promise of strong performance on diverse data while distilling its complexity. Furthermore, our improved model-agnostic distillation strategy can help facilitate interpretability of accurate-but-opaque predictors by choosing a simple understandable model as the student model.

While the majority of enterprise ML applications today involve tabular data and tree models, empirical research on distillation has mostly focused on computer vision applications with only neural network models. Thus, this paper serves a key segment of practitioners that has been overlooked. A major difference in distillation with tabular data are the limited sample sizes of most people's datasets, which means augmentation during distillation is critical. We expect our work to have strong practical impact for these medium/small-scale problems. By allowing practitioners to deploy simpler models that retain the accuracy of their more complex counterparts, our work helps improve the cost of ML inference, the reliability of deployments (student models are less opaque), and may open up new ML applications that were once out of reach due to previously unachievable accuracy-latency limits.

The second avenue for impact is theoretical. The dramatic performance of deep networks on modalities such as images, speech and text has not quite been replicated on tabular data; ensemble methods are still the go-to-methods for such data. One reason for this gap is perhaps that it is difficult to discover invariants for tabular data, in contrast to the pre-baked translation invariance of CNNs for natural images. In the absence of a strong architectural inductive bias, it is important to heavily augment the data to reduce the variance of fitting high-capacity models such as neural networks and handle situations with limited amounts of data. Our work identifies a simple way to achieve this augmentation, where Gibbs sampling is a natural fit that is computationally efficient (because we only need to run a few rounds) and facilitates fine-grained control over the sample-quality vs. the diversity of the augmented samples. Our study of augmentation in the distillation context is different than most existing work on augmentation for supervised learning, where a popular strategy is to use desired invariances that are known a priori to inspire augmentation strategies (since labels are not available for the augmented data in this setting, one typically has to assume each augmented example shares the same label as a real counterpart in the dataset).

**Concerns.** General concerns regarding model distillation include its potential use in "stealing" (cloning) models hidden behind an API. We are not aware of documented occurrences of this practice beyond academic research. This paper does not enhance the capabilities of such attacks as our augmentation strategy to improve distillation requires access to the training data. Another concern is models obtained through distillation may be less reproducible as one needs to repeat both the teacher-training and the student-training exactly. This should be addressed through well-documented code and saving the augmented dataset and all teacher/student/self-attention models to file. A final concern is the role of distillation in model interpretability. Once somebody distills an opaque model into an understandable model that almost retains the average performance of the original model, they may become overconfident that they understand the operating behavior of the opaque model, even though the distilled model may be a poor approximation in certain regions of the feature space (particularly regions poorly represented in the training data due to selection bias). The data augmentation strategy proposed in this paper may actually help mitigate this issue, but is by no means intended to resolve it. For true insight, we recommend careful analysis of the data/models as opposed to the hands-off AutoML + distillation approach presented here.

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
