[Supplementary Material]

# Appendix: Fast, Accurate, and Simple Models for Tabular Data via Augmented Distillation

## A    Methods Details

We not only adopt the AutoGluon[2] predictor as our teacher for distillation, but our experiments also use the AutoGluon implementation of each individual model type (NN, RF, LightGBM, CatBoost) for our student/BASE predictors[3]. Here we consider the same data preprocessing and hyperparameters as AutoGluon uses by default, which have been demonstrated to be highly performant [1].

Unlike the RF/LightGBM/CatBoost models which are implemented in popular third party packages, the NN model is implemented directly in AutoGluon, and offers numerous advantages for tabular data over standard feedforward architectures [1]. This network uses a separate embedding layer for each categorical feature which helps the network separately learn about each of these variables before their representations are blended together by fully-connected layers [59, 63]. The network employs skip connections for improved gradient flow, with both shallow and deep paths connecting the input to the output [54].

Note that all of our student classifiers produce valid predicted probabilities: our neural network student employs a sigmoid output layer to constrain its outputs to $[0, 1]$ in binary classification, and the random forest multiclass students never output negative values (these models do not extrapolate) so we can simply re-normalize their output vectors to have unit-sum.

### A.1    Architecture of our Pseudolikelihood Self-Attention Model

The input layer of our self-attention network applies a linear embedding operation followed by positional encoding. Each internal layer of the network is a Transformer block, which includes two sub-blocks: a multi-head self-attention mechanism and a position-wise fully connected feedforward block [22]. Each of these sub-blocks is wrapped with layer normalization and a residual connection. Here different positions correspond to different features (columns of the table). The output layer of this network produces a mixture of multivariate Gaussians with diagonal covariance, where the final position-wise feedforward block outputs for each feature $i$ both the mean/variance of each Gaussian component ($\mu_k, \sigma_k$) as well as the mixing components ($\lambda_k$). In order to make sure that all input features are on a similar scale, all features are rescaled to mean-zero unit-variance before being fed into our network (and we apply the inverse transform after Gibbs sampling).

Positional encoding is essential for the model to know which value was taken by which feature. For example, without positional encoding: $x^{(1)} = 1, x^{(2)} = 0$ v.s. $x^{(1)} = 0, x^{(2)} = 1$ would lead to similar self-attention input for the third feature $x^{(3)}$ without positional encoding. Thus the representations of our model would suffer, as would its estimated conditional distributions. Here we employ the same sin/cos positional encodings used by Vaswani et al. [22], treating the table column-index of each feature analogously to word positions in a sentence.

Tabular data can contain both numerical and categorical features. In order to have a simple, unified model that can deal with both feature types, we represent categorical features numerically using dequantization [24]. This involves adding uniform noise an the ordered integer encoding of the categories to make these features look numerical to our network. The noise can be inverted via rounding to ensure that discrete categories are produced by our Gibbs sampler (i.e. re-quantization). Dequantization has been successfully employed in a number of deep architectures that otherwise operate on continuous data [62, 65, 67], and allows us to avoid having to employ heterogeneous output layers and unwieldy one-hot enodings.

Table S1 shows our network's hyper-parameters that are used for the experiments in this paper. It is worth noting that we did not conduct any hyper-parameter search to find the best-performing architectures and models. Instead, we simply utilize two different networks: *Small* and *Large*. The *Small* network is used whenever the training dataset has less than $15000$ examples and we otherwise use the *Large* network. Their only differences are in batch sizes and the width of their hidden layers,

all other details such as training procedure, regularization, evaluation protocol, etc. are the same. We utilize two different models in order to avoid overfitting small datasets, and the *Small* network can also be more efficiently trained. We use Adam to optimize the parameters of our network [51].

|  | *Small* | *Large* |
|---|---|---|
| Gaussian mixture components | 100 | 100 |
| Number of layers | 4 | 4 |
| Multi-head attention heads | 8 | 8 |
| Hidden unit size | 32 | 128 |
| Mini-batch size | 16 | 256 |
| Dropout | 0.1 | 0.1 |
| Learning rate | 3E-4 | 3E-4 |
| Weight decay | 1E-6 | 1E-6 |
| Gradient clipping norm | 5 | 5 |

Table S1: Hyper-parameters of our self-attention models.

| Dataset | Type | Sample Size | # Columns | # Classes |
|---|---|---|---|---|
| amazon | binary | 32769 | 9 | - |
| australian | binary | 690 | 14 | - |
| miniboone | binary | 130064 | 50 | - |
| adult | binary | 48842 | 14 | - |
| blood | binary | 748 | 4 | - |
| credit-g | binary | 1000 | 20 | - |
| higgs | binary | 98050 | 28 | - |
| jasmine | binary | 2984 | 144 | - |
| nomao | binary | 34465 | 118 | - |
| numerai28.6 | binary | 96320 | 21 | - |
| phoneme | binary | 5404 | 5 | - |
| sylvine | binary | 5124 | 20 | - |
| covertype | multiclass | 581012 | 54 | 7 |
| helena | multiclass | 65196 | 27 | 100 |
| jannis | multiclass | 83733 | 54 | 4 |
| volkert | multiclass | 58310 | 180 | 10 |
| connect-4 | multiclass | 67557 | 42 | 3 |
| jungle-chess | multiclass | 44819 | 6 | 3 |
| mfeat-factors | multiclass | 2000 | 216 | 10 |
| segment | multiclass | 2310 | 19 | 7 |
| vehicle | multiclass | 846 | 18 | 4 |
| boston | regression | 506 | 13 | - |
| concrete | regression | 1030 | 8 | - |
| energy | regression | 768 | 8 | - |
| kin8nm | regression | 8192 | 8 | - |
| naval | regression | 11934 | 16 | - |
| power | regression | 9568 | 4 | - |
| protein | regression | 45730 | 9 | - |
| wine | regression | 1599 | 11 | - |
| yacht | regression | 308 | 6 | - |

Table S2: Summary of 30 datasets considered in this work, listing the: type of prediction problem, size of the data table, and number of classes for multiclass classification problems. The regression data (along with provided train/test splits) were downloaded from: `https://github.com/yaringal/DropoutUncertaintyExps`. The classification data (with provided train/test splits) were downloaded from: `https://www.openml.org/s/218`. We initially considered additional classification datasets from Gijsbers et al. [39], but decided to not to include those for which: it was trivial to get near 100% accuracy for many model types (so a teacher is unnecessary), the data are dominated by missing values, the original data are extremely high-dimensional ($d > 1000$), or the original data did not come from a table (e.g. Fashion-MNIST).

## B    Experiment Details

We implemented knowledge distillation (KNOW) with classification targets modified as suggested in Hinton et al. [5]. As suggested by Bucilua et al. [4], the distance metric in MUNGE is taken to be the Euclidean distance between (rescaled) numerical features and the Hamming distance between categorical features. Over all datasets, we performed a grid search over MUNGE's user-specified parameters: the feature-resampling probability $p$ and local variance parameter $s$, in order to maximize validation accuracy of the student over $p \in \{0.1, 0.25, 0.5, 0.75\}, s \in \{0.5, 1.0, 5.0\}$. For the conditional tabular GAN, we used the original implementation available at: `https://github.com/sdv-dev/CTGAN`.

On each dataset, we trained AutoGluon for up to 4 hours, and specified the same time-limit for H2O-AutoML and AutoSklearn. When running H2O and AutoSklearn on the 30 datasets, each AutoML tool failed to produce predictions on 2 datasets, and we simply recorded the accuracy/latency achieved by the other tool in this case (such failures are common in AutoML benchmarking, c.f. [1, 39]). Each AutoML tool was run with all default arguments, except for AutoGluon: we additionally set the argument `auto_stack = True` which instructs the system to maximize accuracy at all costs via extensive stack ensembling. We used the same type of AWS EC2 instance (m5.2xlarge) for each predictor to ensure fair comparison of inference times (each tool was run on separate EC2 instance with no other running processes).

For evaluating our Gibbs samples, we computed the Maximum Mean Discrepancy with the mixture-kernel [64], with bandwidths $= [1, 2, 4, 8, 16]$. Our procedure to measure sample fidelity involved the following steps: First we trained our FAST-DAD-Net to maximize pseudolikelihood over data in the training fold. Next we applied Gibbs sampling to generate synthetic samples (initializing the Markov chains at the training data as previously described). Subsequently we assembled a balanced dataset of real (held-out) data from our validation fold which received label $y = 1$ and fake data comprised of Gibbs samples which received label $y = 0$. A random forest was trained on this dataset, and then its accuracy evaluated on another balanced dataset comprised of real data from our test fold (again with label $y = 1$) and fake data comprised of a different set of Gibbs samples (labeled with $y = 0$). The resulting 'sample fidelity' was defined as the distance between this RF accuracy and 0.5.

# C   Additional Results

## (A) Regression

## (B) Binary Classification

## (C) Multiclass Classification

Figure S1: Test accuracy vs latency of individual models and AutoML ensembles, averaged over the: **(A)** regression datasets, **(B)** binary classification datasets, **(C)** multiclass classification datasets. The GIB-1 and BASE dots show performance of *Selected* model (out of the 4 types) on each dataset. Note that for binary classification: the *Selected* BASE models are actually worse than individual RF/LightGBM models, presumably due to overfitting of the validation set via the early-stopping criterion in NN/CatBoost. This issue appears to be mitigated by distillation with augmented data. In multiclass classification, the distilled LightGBM models exhibit worse latency than their BASE counterparts because distillation uses additional data and soft (probabilistic) labels as targets, such that the underlying function to learn becomes more complex. Thus, the depth of its trees grows since LightGBM does not limit it by default. The BASE/distilled latency could easily be matched by restrictively setting the LightGBM depth/leaf-size hyperparameters to ensure equal-sized trees in these two variants.

Table S3: Raw test accuracy (or percent of variation explained $= R^2 \cdot 100$ for regression) under various training/distillation strategies of the *Selected* best individual model (across all 4 model types) chosen based on validation performance. The final column shows the performance of the ensemble-predictor produced by AutoGluon (used as teacher in distillation). Datasets are colored by task: regression (black), binary classification (blue), multiclass classification (red).

| Dataset | BASE | KNOW | MUNGE | HUNGE | GAN | GIB-1 | GIB-5 | GIB-10 | TEACHER |
|---|---|---|---|---|---|---|---|---|---|
| boston | 91.84 | 90.11 | 90.25 | 91.54 | 92.02 | 92.38 | **93.21** | 92.62 | 92.09 |
| concrete | 92.20 | 92.66 | 92.07 | 92.31 | 92.33 | 92.39 | **92.83** | 92.56 | 92.82 |
| energy | 99.86 | 99.85 | 99.91 | 99.92 | 99.87 | **99.93** | 99.92 | 99.92 | 99.93 |
| kin8nm | 93.36 | 93.58 | 94.10 | 93.82 | 94.08 | 93.96 | **94.14** | 94.10 | 93.99 |
| naval | 99.74 | 99.75 | **99.81** | 99.78 | 99.49 | 99.70 | 99.68 | 99.71 | 99.97 |
| power | 96.62 | **96.97** | 96.60 | 96.86 | 96.07 | 96.61 | 96.62 | 96.51 | 97.15 |
| protein | 68.34 | 67.37 | 69.95 | 68.14 | 67.64 | **69.96** | 69.07 | 68.01 | 74.33 |
| wine | 56.44 | 56.53 | 57.38 | 58.61 | 56.42 | **59.27** | 57.80 | 58.49 | 60.74 |
| yacht | 99.24 | 99.55 | 99.88 | 99.92 | 99.93 | **99.94** | 99.94 | 99.90 | 99.87 |
| amazon | 94.84 | 94.81 | 94.72 | 94.87 | 94.69 | **94.90** | 94.81 | 94.81 | 94.96 |
| australian | 86.95 | **88.40** | 85.50 | 85.50 | 85.50 | **88.40** | 86.95 | 85.50 | 86.95 |
| miniboone | 94.50 | 94.71 | 94.77 | 94.40 | 94.44 | **94.86** | 94.44 | 94.64 | 94.88 |
| adult | 87.06 | 87.26 | 87.36 | **87.49** | 86.81 | 86.36 | 86.67 | 86.73 | 87.59 |
| blood | 73.33 | 77.33 | 77.33 | 77.33 | 76.0 | 76.0 | **78.66** | 77.33 | 76.0 |
| credit-g | 71.0 | 78.0 | 78.0 | 76.0 | 75.0 | **80.0** | **80.0** | 77.0 | 79.0 |
| higgs | 72.14 | 73.14 | 73.53 | 72.83 | 72.48 | **73.89** | 73.36 | 73.18 | 73.83 |
| jasmine | **82.94** | 81.93 | 80.26 | 81.93 | 81.93 | 82.27 | 81.27 | 81.93 | 82.60 |
| nomao | **97.30** | 96.98 | 96.72 | 97.15 | 96.98 | 96.77 | 96.83 | 96.86 | 98.20 |
| numerai28.6 | 51.12 | 50.36 | 51.78 | 50.78 | 51.23 | **52.05** | 51.11 | 51.59 | 51.10 |
| phoneme | 89.46 | 90.38 | 90.57 | 90.20 | 90.57 | **91.49** | 90.38 | 90.75 | 92.42 |
| sylvine | 93.56 | 93.37 | 94.15 | 94.34 | 92.39 | 93.56 | 93.95 | **94.54** | 95.32 |
| covertype | 95.90 | 96.99 | **97.00** | 92.84 | 96.39 | 96.19 | 96.48 | 96.06 | 97.66 |
| helena | 38.29 | **40.70** | 40.26 | 39.44 | 39.43 | 40.50 | 39.84 | 40.50 | 40.75 |
| jannis | 70.69 | 72.23 | 72.13 | 70.69 | 71.75 | **72.43** | 72.28 | 71.91 | 73.07 |
| volkert | 69.62 | 71.34 | **72.18** | 69.28 | 70.60 | 71.42 | 70.70 | 70.45 | 74.46 |
| connect-4 | 84.87 | 86.10 | 86.27 | 84.35 | 85.90 | 86.19 | **86.44** | 86.38 | 86.04 |
| jungle-chess | 87.59 | 91.78 | 92.92 | 89.53 | **96.07** | 93.37 | 94.42 | 93.81 | 99.55 |
| mfeat-factors | 98.0 | 97.0 | 97.5 | 97.5 | 98.0 | **98.5** | **98.5** | **98.5** | 98.0 |
| segment | 98.70 | 98.70 | 98.70 | 98.70 | 98.70 | **99.13** | **99.13** | **99.13** | 99.13 |
| vehicle | 83.52 | 77.64 | **88.23** | 87.05 | 83.52 | **88.23** | 87.05 | 87.05 | 85.88 |

Figure S2: Distillation performance when augmented data are: **(A)** additional real data points from the true underlying distribution, **(B)** synthetic examples obtained from 1 round of our Gibbs sampling procedure. Here we report average normalized test accuracy ($R^2$) over the 3 largest regression datasets, with corresponding standard errors indicated by vertical lines (our normalization rescales $R^2$ by the teacher's $R^2$ on each dataset). To obtain additional real data points for augmentation, we did the following: only 20% of the original training set was adopted as the training data (accuracies obtained from this training data shown at 0 on x-axis). The rest of the 80% held-out data was treated as unlabeled and used as augmented data for distillation (in increasing multiples of the training sample-size $n$), following the same distillation procedure described in the main text. The GIB-1 results are obtained by applying our FAST-DAD distillation procedure with only the 20% training data (the 80% held-out data are entirely ignored, so our self-attention pseudolikelihood model is fit to relatively little data). The AutoGluon teacher is also only fit to the same 20% of the training data. Panel **(A)** empirically validates Lemma 1, showing that distillation becomes much more powerful with additional unlabeled data from the true feature distribution. Distillation gains produced by augmenting with Gibbs samples do not match the performance of augmenting with real data, suggesting superior generative models may further reduce this gap.

# D Proof of Theorem 2

Here we discuss our refinement of Lemma 1 that formally describes how the number of steps of Gibbs sampling affects the distillation of the student. Lemma 1 suggests that if we learn a probability distribution $q$ using the data $X_n$, we might be able to reduce the variance term in the VC-bound at the cost of a bias. We now characterize the situation when the Gibbs sampler with a steady-state distribution $q$ is initialized at samples from $p$, namely the original training dataset $X_n$, and is run for $k$ steps. Intuitively, if $k$ is large, the sampler provides data $X'_m$ that is diverse from $X_n$ which leads to stronger variance reduction. However it is also true that the samples $X'_m$ are not drawn from $p$ and therefore the teacher $f$ suffers a covariate shift on these samples which leads to poor fitting of the student $g$. This suggests there should be a sweet spot: the number of Gibbs sampling steps $k$ should lead to variance reduction but should not be so large as to cause a large covariate-shift/bias. We capture this phenomenon in the following theorem. For simplicity, we only consider the special case where $m = n$. Our proof can be generalized to $m \neq n$ but the details of the underlying symmetrization argument are more intricate (see comments in the proof). We stick to this special case to elucidate the main point. The full theorem statement is repeated here for completeness.

**Theorem 2 (Refinement of Lemma 1)** *Under the assumptions of Lemma 1, suppose that the student $g^*$ is chosen to minimize $D_{\mathrm{emp}}(f, g, X_n \cup X'_n)$ where $X'_n$ are $n$ samples drawn after running the Gibbs sampler initialized at samples from $X_n$ for k-steps. Then there exist constants $V, c$ and $\delta > 0$ such that with probability at least $1 - \delta$ we have*

$$D(f, g^*, p) \leqslant D_{\mathrm{emp}}(f, g^*, X_n \cup X'_n) + \sqrt{\frac{4V(c + \Delta_k) - \log \delta}{n}} + \Delta_k. \tag{6}$$

*The quantity $\Delta_k = \|T_q^k p - p\|_{\mathrm{TV}}$ is the total-variation distance between the true data distribution $p$ and the distribution of the sampler's iterates after $k$ steps, denoted by $T_q^k p$. The steady-state distribution of the Gibbs sampler is denoted by $q$.*

**Proof** Let $q$ be the steady-state distribution of the Gibbs sampler with a linear operator $T_q$ that denotes the one-step transition kernel. Under general conditions [66], the distribution of the iterates of the sampler converges to this steady-state distribution as $k \to \infty$, i.e.,

$$\lim_{k \to \infty} T_q^k \nu = q,$$

from any initial distribution $\nu$. Explicit rates are available for this convergence [33]: there exist constants $\lambda \in (0, 1)$ and $c(q)$ such that

$$\|T_q^k p - q\|_{\mathrm{TV}} \leqslant c(q) \lambda^k. \tag{7}$$

where $\|\nu - \mu\|_{\mathrm{TV}} := 2 \sup \{|\nu(A) - \mu(A)| : A \in \mathcal{B}(\mathcal{X})\}$ denotes the total-variation norm; the set $\mathcal{B}(\mathcal{X})$ is the Borel $\sigma$-algebra of the domain $\mathcal{X}$. These rates are sharp for some parametric models [56]. We use the following shorthand to denote, $T_q^k p$, the density obtained after applying the one-step transition kernel $k$ times.

$$q^k := T_q^k p.$$

Suppose that the Gibbs sampler initialized at $p$ runs for $k$ steps and we then sample a dataset $X'_n$ of $n$ samples from the resultant distribution $T_q^k p$:

$$X'_n = \left\{ x'_i \sim T_q^k p \right\}_{i=1,\dots,n}.$$

The samples in $X'_n$ are correlated with those already in $X_n$. The student is fit to this dataset $X_n \cup X'_n$ where the samples are not independent (we don't have $X_n \ni x \perp\!\!\!\perp x' \in X'_n$) or identically distributed ($x \sim p$ and $x' \sim T_q^k p$). Characterizing generalization performance is difficult for this scenario and requires strong assumptions, c.f. [55], but we can we make the following helpful simplification.

**Assumption 1** *The number of Gibbs steps $k$ is large enough for the samples in $X_n$ and $X'_n$ to be statistically independent.*

Note that this does not imply that the samples are identically distributed, they still come from distributions $p$ and $T_q^k p$ respectively. Since $k$ is the product of the number of rounds of Gibbs sampling and the dimensionality of the data ($d$), achieving approximate independence does not necessarily require a large number of Gibbs rounds.

We now employ a bound by Jonathan Baxter [32] that studies the generalization performance of a model $g$ when it sees data from a mixture of two different, possibly correlated, distributions, $p$ and $q^k$. This is a uniform-convergence bound and follows via a two-step symmetrization argument where the second step involves separate permutations of the samples in datasets $X_n$ and $X_n'$. The same technique as that of [32] also works if we draw more data $X_m'$ from the new distribution than the original dataset $X_n$, i.e., if $m \geqslant n$. However the details are intricate and we stick to this special $m = n$ case to elucidate the main point.

For all functions $g \in \mathcal{G}$, in particular for $g^* = \mathrm{argmin}_g D_{\mathrm{emp}}(f, g, X_n \cup X_n')$, the following holds with probability at least $1 - \delta$:

$$D\left(f, g, \frac{p + q^k}{2}\right) \leqslant D_{\mathrm{emp}}(f, g, X_n \cup X_n') + \epsilon$$

$$\text{if} \qquad n \geqslant \frac{c}{\epsilon^2} \log \frac{N(\epsilon, \mathcal{G}^2)}{\delta} \tag{8}$$

where $c$ is a constant. The quantity $N(\epsilon, \mathcal{H})$ is the $\epsilon$-net covering number of the hypothesis class $\mathcal{H}$ under a given metric $m$ [32]. According to Baxter's result, for our case with two tasks, $p$ and $q^k$, we are interested in computing the covering number for $\mathcal{H} = \mathcal{G} \times \mathcal{G}$ and the metric $m$ between two functions in $g, g' \in \mathcal{G}^2$ as

$$m(g, g') = \frac{1}{2} \int \left| d(g(x_1), f(x_1)) + d(g(x_2), f(x_2)) - d(g'(x_1), f(x_1)) - d(g'(x_2), f(x_2)) \right| \, \mathrm{d}p(x_1) \, \mathrm{d}q^k(x_2).$$

with the labels $f(x_i)$ given by the teacher. Our hypothesis class $\mathcal{H}$ on the two tasks is the Cartesian product of the hypothesis class $\mathcal{G}$. Haussler's theorem [60] gives an upper bound on the covering number in terms of the VC-dimension

$$\log N(\epsilon, \mathcal{H}) \leqslant 2 V_{\mathcal{H}} \log(c/\epsilon). \tag{9}$$

where $V_{\mathcal{H}}$ is the VC-dimension [68] of $\mathcal{H}$, $c > 1$ is a constant, and $\log N(\epsilon, \mathcal{H})$ is also called the metric entropy.

Observe that the left-hand side in (8) can be written as

$$D\left(f, g, \frac{p + q^k}{2}\right) = D(f, g, p) + \frac{1}{2} \int d(f(x), g(x)) \, \mathrm{d}(q^k - p). \tag{10}$$

Let us define

$$\Delta_k := \|T_q^k p - p\|_{\mathrm{TV}}.$$

We next analyze the metric $m(g, g')$ where we note that again $q^k = T_q^k p$.

$$m(g, g') \leqslant \int \left| d(g(x), f(x)) - d(g'(x), f(x)) \right| \, \mathrm{d}p(x)$$

$$+ \frac{1}{2} \int \left| d(g(x), f(x)) - d(g'(x), f(x)) \right| \, \mathrm{d}\left(T_q^k - p\right)(x)$$

$$\leqslant \int \left| d(g(x), f(x)) - d(g'(x), f(x)) \right| \, \mathrm{d}p(x) + \|T_q^k p - p\|_{\mathrm{TV}}.$$

Similarly we also have

$$\int \left| d(g(x), f(x)) - d(g'(x), f(x)) \right| \, \mathrm{d}p(x) - \|T_q^k p - p\|_{\mathrm{TV}} \leqslant m(g, g').$$

In other words, the distance $m(g, g')$ on the Cartesian space $\mathcal{G}^2$ can be upper bounded by the distance between $g, g'$ on the original space $\mathcal{G}$ up to an additive term $\Delta_k$ that increases with the number of steps $k$ of Gibbs sampling.

Next observe that we have an upper bound on the metric entropy

$$\log N(\epsilon, \mathcal{G}^2) \leqslant 2 \log N(\epsilon, \mathcal{G}) \tag{11}$$

if the two datasets $X_n$ and $X'_n$ are iid. If the datasets are not iid, using the calculation for $m(g, g')$ above, computing the size of the $\epsilon$-net for $\mathcal{G}^2$ is effectively the same as changing $\epsilon$ on the right hand side of (11) to

$$\epsilon' = \epsilon - \Delta_k$$
$$= \epsilon \left(1 - \frac{\Delta_k}{\epsilon}\right).$$

Plugging the previous two expressions into (9) implies

$$\log N(\epsilon, \mathcal{G}^2) \leqslant 2 \log N(\epsilon, \mathcal{G})$$
$$\approx 4 V_{\mathcal{G}} \log \left(\frac{c}{\epsilon}\left(1 + \frac{\Delta_k}{\epsilon}\right)\right)$$
$$= 4 V_{\mathcal{G}} \left(\log(c/\epsilon) + \frac{\Delta_k}{\epsilon}\right) + o((\Delta_k/\epsilon)^2). \tag{12}$$

The approximation above is valid if we additionally assume $\Delta_k \ll \epsilon$. We have thus shown that there exists a constant $V$ such that with probability at least $1 - \delta$:

$$D(f, g^*, p) \leqslant D_{\text{emp}}(f, g^*, X_n \cup X'_n) + \sqrt{\frac{4V(c + \Delta_k) - \log \delta}{n}} + \frac{1}{2}\int d(f(x), g^*(x)) \, \mathrm{d}(p - q^k)$$
$$\leqslant D_{\text{emp}}(f, g^*, X_n \cup X'_n) + \sqrt{\frac{4V(c + \Delta_k) - \log \delta}{n}} + \Delta_k \tag{13}$$

The inequality follows because

$$\left|\int d(f(x), g^*(x)) \, \mathrm{d}(p - q^k)\right| \leqslant \left|\int \mathrm{d}(p - q^k)\right| \leqslant 2\|p - q^k\|_{\text{TV}}$$

since $d(\cdot, \cdot) \leqslant 1$. Recall $k$ is the number of Gibbs sampling steps, $c$ is a constant, and $g^* = \underset{g}{\text{argmin}} \ D_{\text{emp}}(f, g, X_n \cup X'_n)$. ∎

We provide some additional comments on this result. Note that $\Delta_k \to \|p - q\|_{\text{TV}}$ as $k \to \infty$, so it increases with the number of Gibbs sampling steps $k$. We can draw a large number of samples $n$ from $q^k$ to reduce the second term in the bound. Using a large $k$ is both computationally inefficient and may also cause a bias given by the additive term of $\Delta_k$ (third term), if the stationary distribution $q$ of our Gibbs sampler poorly approximates $p$. As our pseudolikelihood model is fit to limited data in practice, it is thus better to draw a large number of samples from earlier steps, i.e. using only a few steps of Gibbs sampling from each training datum instead of running a long chain. Among all $k$ that produce samples which are approximately independent of the original training data, we would like to use the smallest.

The experiments in our paper empirically show that, on an average over many datasets, running the Gibbs sampler for 1–5 rounds (one round involves performing a Gibbs step for every conditional in the pseudolikelihood) works better than running it for longer. Note that if we employ fewer steps than even a single round of Gibbs sampling, the augmented data will be highly dependent on the training data as some features will not have been resampled, thus diminishing the *effective sample size* of the student's distillation dataset. It is also readily seen from the above bound that if the Gibbs sampler is initialized at a distribution other than $p$, we would need a large number of steps $k$ before the bias term $\|T_q^k \nu - p\|_{\text{TV}}$ is adequately small.

## Additional References for the Appendix

[32] J. Baxter. A model of inductive bias learning. *Journal of artificial intelligence research*, 12: 149–198, 2000.

[4] C. Bucilua, R. Caruana, and A. Niculescu-Mizil. Model compression. In *Proceedings of the 12th ACM SIGKDD international conference on Knowledge discovery and data mining*, pages 535–541, 2006.

[54] H.-T. Cheng, L. Koc, J. Harmsen, T. Shaked, T. Chandra, H. Aradhye, G. Anderson, G. Corrado, W. Chai, M. Ispir, et al. Wide & deep learning for recommender systems. In *Proceedings of the 1st workshop on deep learning for recommender systems*, pages 7–10, 2016.

[55] Y. Dagan, C. Daskalakis, N. Dikkala, and S. Jayanti. Learning from weakly dependent data under Dobrushin's condition. *arXiv preprint arXiv:1906.09247*, 2019.

[56] P. Diaconis, K. Khare, and L. Saloff-Coste. Gibbs sampling, conjugate priors and coupling. *Sankhya A*, 72(1):136–169, 2010.

[1] N. Erickson, J. Mueller, A. Shirkov, H. Zhang, P. Larroy, M. Li, and A. Smola. AutoGluon-Tabular: Robust and accurate AutoML for structured data. *arXiv preprint arXiv:2003.06505*, 2020.

[39] P. Gijsbers, E. LeDell, J. Thomas, S. Poirier, B. Bischl, and J. Vanschoren. An open source AutoML benchmark. In *ICML Workshop on Automated Machine Learning*, 2019.

[59] C. Guo and F. Berkhahn. Entity embeddings of categorical variables. *arXiv preprint arXiv:1604.06737*, 2016.

[60] D. Haussler. *Probably approximately correct learning*. University of California, Santa Cruz, Computer Research Laboratory, 1990.

[5] G. Hinton, O. Vinyals, and J. Dean. Distilling the knowledge in a neural network. *NIPS Deep Learning and Representation Learning Workshop*, 2015.

[62] E. Hoogeboom, T. S. Cohen, and J. M. Tomczak. Learning discrete distributions by dequantization. *arXiv preprint arXiv:2001.11235*, 2020.

[63] J. Howard and S. Gugger. fastai: A layered api for deep learning. *arXiv preprint arXiv:2002.04688*, 2020.

[64] C.-L. Li, W.-C. Chang, Y. Cheng, Y. Yang, and B. Póczos. Mmd gan: Towards deeper understanding of moment matching network. In *Advances in Neural Information Processing Systems*, pages 2203–2213, 2017.

[65] X. Ma, X. Kong, S. Zhang, and E. Hovy. Macow: Masked convolutional generative flow. In *Advances in Neural Information Processing Systems*, pages 5891–5900, 2019.

[66] C. Robert and G. Casella. *Monte Carlo statistical methods*. Springer Science & Business Media, 2013.

[67] L. Theis, A. van den Oord, and M. Bethge. A note on the evaluation of generative models. In *International Conference on Learning Representations*, 2016.

[68] V. N. Vapnik and A. Y. Chervonenkis. On the uniform convergence of relative frequencies of events to their probabilities. In *Measures of complexity*, pages 11–30. Springer, 2015.

[22] A. Vaswani, N. Shazeer, N. Parmar, J. Uszkoreit, L. Jones, A. N. Gomez, L. Kaiser, and I. Polosukhin. Attention is all you need. In *Advances in Neural Information Processing Systems*, 2017.

[33] N.-Y. Wang, L. Wu, et al. Convergence rate and concentration inequalities for gibbs sampling in high dimension. *Bernoulli*, 20(4):1698–1716, 2014.

## Footnotes

[2] https://github.com/awslabs/autogluon/

[3] Although we selected AutoGluon as the AutoML tool for this paper's experiments, we emphasize that none of our distillation methodology is specific to AutoGluon teachers/students.