[Reviews · NeurIPS 2020]

Review 1

Summary and Contributions: Thanks for the responses. From discussions with fellow reviewers, I think it would be worth at least an extra sentence or two expanding on the fact that (or at least in my understanding) that as you run more rounds of Gibbs sampling, your sampling distribution is moving way from the true data-generating distribution towards your learned approximation. Still, it's not obvious that this is bad, since you will also attain artificial training examples that are more different from your original training set, which may be helpful. And this is why you didn't know a priori how many rounds of Gibbs sampling to run. Also, as I suggest below, it might even be helpful to do a partial round of Gibbs sampling. --- Today's best off-the-shelf performance on Tabular datasets is provided by AutoML solutions, such as AutoGluon, AutoSklearn, and H20-AutoML. The models produced are very large ensembles and can be orders of magnitude slower than individual models at prediction time. One approach to attain a simpler and faster model is model distillation, in which a new model (the student) is trained to produce the predictions of the big model (the teacher). Typically, the resulting model is faster but has worse performance. The motivating idea of this paper is that a significant portion of the performance drop is due to "estimation error", which is the gap between the student and the teacher due to only having a limited amount of training data. The proposal of the paper is to model the input distribution and to use Gibbs sampling to generate generate artificial inputs, which are then labeled by the teacher model and used to augment the training set. Their distillation method is compared to several other methods on 30 machine learning problems. They generally achieve significant speedup with moderate performance loss.

Strengths: Having a single transformer-like model that learns all feature conditional distributions at once is a nice contribution. The idea of using these models together with Gibbs sampling for generating more unlabeled data is new to me, though I'm not sure it's new. In truth, I don't think generating unlabeled tabular data is a common task. Various intuitions are provided for the tradeoffs of running more rounds of Gibbs sampling, and these are supported by various theorems. Work seems quite relevant to neurips community.

Weaknesses: I would have liked to see more discussion on what can go wrong with the Gibbs sampling part. For example, doesn't Gibbs sampling get stuck when two features are highly correlated? With GIB-1 often performing best, did you try running for fractions of a round?

Correctness: Didn't notice any issues.

Clarity: Generally yes. But here's something that wasn't clear to me: I thought you're predicting the conditional distribution for one feature given the rest of the features. Why then would you be producing a _multivariate_ mixture of Gaussians? Why not just a univarate mixture of Gaussians?

Relation to Prior Work: Yes

Reproducibility: Yes

Additional Feedback: Line 42: "statistical approximation-error" -- The error due to having a limited training set is usually referred to as "estimation error", while "approximation error" usually refers to the component of the error that's due to limitations in the expressivity of your hypothesis space (i.e the models you're selecting from during training). Line 146-147: You make a claim that "maximum pseudolikelihood estimation produces asymptotically consistent parameter estimates" and compare it to optimizing the likelihood. Consistent parameter estimates for what? For a joint distribution on x? It might help if you are more explicit about what the likelihood approach would be that presumably the pseudolikelihood method is approximating.


Review 2

Summary and Contributions: The paper describes a method for model distillation adapted to tabular, limited-size datasets. To this end, it develops a data augmentation technique based on Gibbs sampling, pseudo-likelihood and self-attention, so that student models are exposed to sufficient data to avoid overfitting. Both distillation and augmentation method are motivated theoretically, by analytic experiments and functional experiments on many different, classic benchmark datasets.

Strengths: I totally relate to the motivation for this work in my professional practice, and expect this to be widely useful. The best that could happen to this method is to become unnoticeable within a broader set of AutoML techniques. This work is novel and different from related work due to its encompassing several types of popular models (as opposed to just one in predecessor work) and tasks. It is relevant to the Neurips community because it brings a solution to a practically important problem through technically sophisticated means, with good theoretical grounding. The methods employed are sound and non-trivial. For instance it looks like a good idea to focus on density estimation of conditionals for data augmentation, and the means adopted here (PL+self attention model) are elaborate. Reproducibility I expect to be quite good, code is shared, many details are given. I have not tried run the code; a full result reproduction script including random seeds does not seem to be provided. I agree with the points raised in the Broader Impact Statement, especially on the lack of methods targeting tabular data for both distillation (line 345 sqq) and augmentation (line 363 sqq).

Weaknesses: I find it concerning that the best performing number of Gibbs rounds is 1, addressed line 267 sq and 656 sqq. This is visible in table 1, results for GIB-1 through -5. This makes fig 3a, which goes up to GIB-200, mostly irrelevant for the purpose of distillation. It puts a shadow on the design of the self-attention network constructed to model pseudo-likelihood: too large? too many parameters? wrong design? Differences in table 1 and fig 1 are very small, I can't convince myself they are significant. A partial indication is given by p-values in table 1 (how are they calculated?)

Correctness: Yes. The evaluation is fine. I could find no unconvincing statement in the paper. The proofs seem complete and correct.

Clarity: Yes. The paper is outstandingly clear. The writing is terse and to the point. The choice of structure is good.

Relation to Prior Work: Yes. The closest work seems to be ref 4. The extensive citation of relevant references is a (secondary) strength of the paper.

Reproducibility: Yes

Additional Feedback: # Suggestions - clarify what the setting "auto_stack" achieves line 241 - clarify what is meant by "Selected" model in the captions of figures by a ref to sec4 - fig 3b: horizontal axis legend: specify "% absolute improvement over BASE" - table 1 caption: specify that column rank is obtained as average over datasets (I guess?) # Minor errors - ref 24 and 30 must be completed - 224 c.f. - 180 onto -> on to - 290 fig 3bA - 293 distillation period ?? - 364 different than ---- note after reviewer discussion I have carefully read the author feedback, other reviews, and the reviewer discussion. I'm happy to maintain my score. One point should be discussed better: why does the paper insist on analysing Gibbs sampling behaviour until iteration 100, if experimental evidence shows that only the first brings good quality samples? In discusion, reviewers have concluded that since the conditional distributions are imperfect, further sampling takes us away from the stationary distribution. Do the authors agree with this?


Review 3

Summary and Contributions: This paper focuses on the problem of obtaining accurate models for tabular data that are efficient (i.e., low latency) during deployment. The authors propose using AutoML approaches to first learn an accurate but expensive to deploy network and then opt to use distillation to obtain a smaller and more efficient network. The core technical problem that this work addresses is how to mitigate the accuracy drop after distillation. This work introduces a data augmentation approach for tabular data that relies on an attention-based estimator and Gibbs sampling. After a set of artificial training points are obtained a student network is trained over the outputs of the teacher network and the original training data. This approach is shown to perform favorably compared to other distillation approaches.

Strengths: This paper puts forward an interesting and sound approach for addressing the problems of distillation for models over tabular data. While the individual technical contributions of this work build heavily on prior works (please see the weak points next) their combination to design a solution mitigates the accuracy drop of distilled networks is itself novel and very practical. In addition, as demonstrated by the experimental results in the paper the approach is applicable to different AutoML frameworks; Generality is a strong point. Finally, the authors consider an extensive array of distillation methods and AutoML frameworks.

Weaknesses: Despite the strengths outlined above there are three main weakness in this work. 1) The current manuscript is missing a comparison to approaches that do not rely on distillation to obtain efficient-to-deploy networks; 2) In several places the presentation omits details that in turn raise questions on the setting considered in this work; and 3) the novelty of individual components of the proposed solutions is limited. 1. Comparison to non-distillation methods. While I was reading this paper I kept wondering, why is distillation the only approach to obtain an efficient model given the output of an AutoML framework? There are different methods in the literature that aim to solve the same problem of getting efficient to deploy models: For example, learnable model cascades (e.g. Willump from MLsys'20 https://mlsys.org/Conferences/2020/Schedule?showEvent=1416) aim to solve a similar problem. The authors do not discuss any such works and do not compare against systems like Wilump, hence, I was left with the fundamental question: why is distillation better and how does it compare to model cascades? Given the above, this work felt that it fast focused on the technical challenges of the niche solution around distillation and forgot the bigger question posed at the beginning of the introduction. 2. Questions/Issues with the Presentation: From the Intro, Figure 1: The legend is not easy to interpret since it follows two different abstractions: are colors versus the pointer type supposed to indicate two different dimensions of the evaluated models? I am not sure what the Green cross, diamond etc indicate, are those distilled models, and from which automl system were they obtained? Moreover, I am rather skeptical seeing only the mean. I would have loved to understand where your methods is significantly better and when does it fail, like a best-case, worst-case, average-case analysis. Reporting the mean alone can be misleading. In Section 3.1 (Maximum Pseudo-likelihood Estimation) Tabular data typically contains numerical, categorical, and text-based data. How do you handle mixed-type and hence mixed distribution tabular data? I have a hard time seeing how a mixture of Gaussians is a good distribution for modeling mixed-type data. Assumptions on the input data should be stated clearly. In line 165-166, why is the fact that your approach does not compute all conditionals simultaneously beneficial for the task you are considering? This comparison to prior work seems obscure as it does not provide any intuition as to how the two approaches compare with respect to pseudo likelihood estimation. Section 3.2 line 179: is Gibbs sampling performed over all samples in the training data or per sample. It is not clear if every feature sampling operates at the column/attribute-level or at a per sample approach. If the latter, do you assume that all samples are iid? Also, in line 179 does x^{-I} contain the resampled values or the original, observed values? Section 3.2: like 193 - 197 (accurate p/many rounds of Gibbs vs inaccurate p/early stopping): I am wondering if there is an operational procedure for one to identify in which of the two regimes the learned estimator lies and hence optimize how many rounds of Gibbs sampling are needed. In other words, what are the conditions that yield accurate conditional probability estimates? I suspect it is related to the inherent structure of the tabular data and the dependencies between features. A discussion of this points would greatly improve the manuscript, given that the main strength of this work is its general applicability. Line 221 with the reference to Bucilua et al.: What is the alternative multiclass-strategy in your binary classification experiments? Please provide a forward reference if applicable. Experimental evaluation: Apart from the missing comparison to non-distillation methods, the authors need to perform a pass and fix some minor issues in presentation. Figure 3: What are the units of the y-axis? Also when you refer to percentage of points in Fig 3b, this is points of what? Table 1: indicates that most of the time one round of Gibbs sampling is enough. Does this mean that your estimates of the pseudo-likelihood are not accurate, given your argument in 193-197? Why is it the case that if one uses more steps (10) then the performance of your approach is comparable to baselines? Figure 4: the same criticism for its legend as with Figure 1. C. Novelty Section 3.1: Your idea of using self-attention to model tabular data is not new and has been already explored by several recent works that introduce highly relevant models (please see the references in the Relation to Prior Work part of the review). Distillation for efficient models is also not novel. Given this, please provide a better description as to what is the main technical novelty of this work.

Correctness: The proposed approaches seem correct. There are some details that are not clear as described in the weaknesses above.

Clarity: The paper is mostly well written but there are several details that are missing from the presentation (See weaknesses).

Relation to Prior Work: As mentioned above I would encourage the authors to try and address the bigger question they pose in the intro, i.e., how can one make AutoML models efficient and achieve low-latency predictions. As such the authors need to extend their discussion and comparisons to methods from model cascades. In addition, the authors omit to discuss how their self-attention model compares to prior works that leverage attention-based models to capture the distribution of tabular data. Some example are: TabNet https://arxiv.org/abs/1908.07442 (although it focuses mainly on predictive models and not estimators) and the attention model from Wu et al, MLSys 2020 (https://proceedings.mlsys.org/papers/2020/123) which captures the distribution of mixed-type tabular data using a masked self-attention as mechanism (admittedly simpler than) yours. You should at least discuss these works.

Reproducibility: Yes

Additional Feedback: After Author response: Most of my concerns have been addressed. There is one point the authors should still address: the behavior of Gibbs sampling. Based on what I am reading in line 132, the sampling approach if restricted to one round, it corresponds to performing sampling over the conditional given the initial observations per sample (tuple in the tabular dataset). Given this and the empirical observations in the paper it seems that Gibbs is not offering anything. So why not present this as a negative result but present it as a contribution? It seems that one round of GIbbs gives you an augmented data set by introducing variations that are close to the original distribution without introducing bias. The exps show that augmentation constrained to bounded distr. distance helps, but then multiple rounds of Gibbs leads to increased distr distance and hence augmentation using the “shifted” distribution hurts performance. You should discuss this further.


Review 4

Summary and Contributions: This paper proposes a new knowledge distillation algorithm for tabular data. The proposed algorithm firstly learn the data conditionals p(x_i | x_{-i}) with a transformer model, and then train the distilled model with augmented training set obtained by Gibbs sampling. The authors also give a generalization bound of the proposed approach. State-of-the-art performance is obtained on a wide range of datasets.

Strengths: The proposed algorithm is technically sound, and also has good theoretical grounding (generalization bound). Empirical result is good. Tabular data are pervasive and important for industrial applications such as advertisement and finance, and the proposed approach may have real impact in these areas.

Weaknesses: I am not sure if the proposed transformer + Gibbs sampling approach is necessary and idiomatic. Why don't we directly learn an autoregressive transformer p(x_i | x_{<i}), and directly sample from these model distribution, without performing Gibbs sampling at all? This simplify both the implementation and the theory. I think this needs to be better justified. Furthermore, the main contribution of this paper seems to be the data augmentation algorithm. I would like to see more direct empirical study of the data augmentation algorithm itself (such as the distance between p and q in the theorems). Post rebuttal ==== I have read the author response. Since my concerns are not addressed, I will leave my score unchanged.

Correctness: Claims, method, and empirical methodology all seem to be correct.

Clarity: I think this paper is very well written.

Relation to Prior Work: I am not an expert in the field of autoML and distillation. The difference between the proposed approach with MUNGE might be better described.

Reproducibility: Yes

Additional Feedback:

[Author Response · NeurIPS 2020]

We thank reviewers for thoughtful suggestions which we'll add to improve our paper and are glad they find this work
**high-impact** & **novel**. Abbrev: neural net (*NN*), gradient boosting (*GBM*), random forest (*RF*), pseudolikelihood (*PL*).

**R2+R3:** *Quality of PL estimates.* From limited data, it's *impossible* for *any* method to accurately estimate nonparametric
multivariate distribution. Even when estimator is *inaccurate*, few Gibbs rounds initialized at data enable good distillation
via our approach. This is subject of Thm 2 & Fig 2 (also shows mixed samples from PL-estimator can be high-quality
for complex $p$ with big $n$). Using these PL-estimates, our method achieves **1000×** **speedup** with minimal accuracy-loss
across **30 diverse datasets**. GIB-1,5,10 all greatly improve BASE models.

**R1.** *Sampling correlated features?* We'll clarify Gibbs' mixing rate may slow for highly-correlated features (Wang
et al 2014 [32]). We anyway do *not* run Gibbs sampling till mixing as its stationary distribution is an approximation
(learned from limited data). Fig 2 & 3a-*Diffusion* (and overall distillation performance) show this isn't a practical issue.
*Consistency of maximum pseudolikelihood.* We'll clarify consistency is *not* referring to our self-attention model, but
rather generally: For many *parametric* models (assuming parametric underlying joint distribution), maximum PL
produces consistent parameter estimates (Besag 1997 [23]), which implies consistency of estimated joint distribution.
As suggested by R1, we'll clarify: (1) per-feature conditional distribution is just *univariate* mixture of Gaussians, not
multivariate. (2) L24 is about estimation not approximation error. (3) We didn't try fractions of Gibbs-round to avoid
dimensionality-dependent hyperparameter, but this may boost practical performance.

**R2.** *Table 1 p-values?* These are computed via one-sided paired $t$-test of method-performance vs BASE-performance
on each dataset (datasets = observations). Differences should be statistically significant where $p < 0.05$.
As suggested by R2, we'll fix typos issues, use clearer captions, and clarify: (1) In Table 1: Rank is computed by
ranking methods' performance from 1-9 on each dataset, and computing average of these ranks (over datasets).
(2) `auto_stack` activates automated stack ensembling in AutoGluon which boosts accuracy but harms latency.
(3) Fig 3a has GIB-200 to show sampler's stationary behavior (estimated from limited data, stationary $q$ is inaccurate).

**R3.** *Why Distillation?* We study how to improve upon latency of ensemble predictors while preserving their accuracy.
Our distillation strategy produces 1000× speedup and models that are faster & more accurate than other AutoML
over 30 datasets. Thus it is practically performant, and it is very *broadly* applicable. We'll clarify distillation is *not*
only way to improve latency and cite cascades & NN-compression (Willump). Distillation can be applied to *any*
AutoML tool's ensemble and *any* student model type (may be important if user has particular inference-accelerator /
hardware-constraints). Cascades instead require *modifying* an existing AutoML system, and are more complex to deploy
than our single distilled models. We don't know any accurate AutoML system for *tabular* data that offers cascades.
*Non-distillation approaches to obtain efficient-to-deploy networks?* Note we're *not* compressing a large NN model
(NN models for tabular data tend to be quite small as NN-accuracy quickly plateaus with size), but a *more-accurate*
heterogeneous model-ensemble of NN + other models. Many NN-compression approaches are not appropriate for
heterogeneous ensembles, and are limited to NN-student models (unlike distillation).
*Main technical novelty?* We did not claim distillation nor tabular self-attention are novel in this work. Main novelty
is our overall distillation strategy (and theoretical insights about it), which is quite different than previous works
by combining 4 ideas: 1) augmenting student's training set, 2) generative model for augmentation that estimates
just conditionals, not joint distribution, 3) one model estimating all conditionals via PL, 4) augmentation via Gibbs
sampling warm-started at datapoints themselves, which facilitates control (unlike say GAN). Plus, we present first *large*
distillation benchmark with 30 regression+classification tabular datasets, and many student models (GBM/RF/NN).
*Metrics besides mean performance.* Table 1 reports $p$-values (to account for spread), broken down for each problem-type.
Fig 3b reports *median* + interquartile range (to show spread). Raw performance on each dataset is listed in Table S3.
*Handle mixed-types?* Dequantization is applied, see Appendix A.1. More sophisticated approaches left for future work.
*Line 165-166?* We'll clarify masked self-attention allows us to use *one* set of parameters to model all conditionals,
rather than $d$ models (one for each conditional, which is cumbersome), as mixture density networks would require.

As suggested by R3, we'll clarify: (1) TabNet & Wu et al in related work (as R3 says they only consider small pieces of
our overall task: Wu et al only model conditionals, TabNet is just a predictive model). (2) Fig 3a y-axis is unitless as
each measure has been normalized to [0,1], and *percentage points* in Fig 3b = distilled student's accuracy minus BASE's
accuracy. (3) Add forward ref to Sec 4 in Fig 1 legend that explains color = overall training strategy, star/plus/X/diamond
= type of single student/BASE model. (4) We do *not* distill binary classifications problems as if they were multiclass
classification problems (which would use log-loss instead of our Brier score). (5) As in standard Gibbs sampling, $x^{-i}$ is
updated with value sampled in previous step (sampling just one column per step), and we initialize separate sampler
with every training datapoint (duplicating some initial points to create $m > n$ augmented datapoints). (6) One could
potentially use (held-out) pseudolikelihood-estimates' performance to adaptively select number of Gibbs rounds.

**R4.** *Consider $p(x_i|x_{<i})$?* L167 states: autoregressive models are undesirably sensitive to *order* of columns.
*Empirical study of augmentation algorithm?* Fig 2 shows qualitative evaluation of Gibbs-augmentation. Fig 3a studies
sample-diversity (*Diffusion*,*Discrepancy*) vs. distance between $p$ and $q$ (*Fidelity*,*Discrepancy*) in Gibbs-augmentation.

[Meta-Review · NeurIPS 2020]

Four knowledgeable referees support acceptance for the contribution mainly due to its novelty and practical impact, and I also recommend acceptance. However, please consider adding more discussions about the effect of multiple Gibbs sampling rounds and about the need and justification of Gibbs sampler, to address reviewers questions/concerns after rebuttal.